

# Contrast optimization of mass spectrometry imaging (MSI) data visualization by threshold intensity quantization (TrIQ)

Ignacio Rosas-Román and Robert Winkler

Biotechnology and Biochemistry, CINVESTAV Unidad Irapuato, Irapuato, Guanajuato, Mexico

## ABSTRACT

Mass spectrometry imaging (MSI) enables the unbiased characterization of surfaces with respect to their chemical composition. In biological MSI, zones with differential mass profiles hint towards localized physiological processes, such as the tissue-specific accumulation of secondary metabolites, or diseases, such as cancer. Thus, the efficient discovery of 'regions of interest' (ROI) is of utmost importance in MSI. However, often the discovery of ROIs is hampered by high background noise and artifact signals. Especially in ambient ionization MSI, unmasking biologically relevant information from crude data sets is challenging. Therefore, we implemented a Threshold Intensity Quantization (TrIQ) algorithm for augmenting the contrast in MSI data visualizations. The simple algorithm reduces the impact of extreme values ('outliers') and rescales the dynamic range of mass signals. We provide an R script for post-processing MSI data in the imzML community format (https://bitbucket.org/lababi/msi.r) and implemented the TrIQ in our open-source imaging software RmsiGUI (https://bitbucket.org/lababi/rmsigui/). Applying these programs to different biological MSI data sets demonstrated the universal applicability of TrIQ for improving the contrast in the MSI data visualization. We show that TrIQ improves a subsequent detection of ROIs by sectioning. In addition, the adjustment of the dynamic signal intensity range makes MSI data sets comparable.

Corresponding author
Robert Winkler,
robert.winkler@cinvestav.mx

# INTRODUCTION

Imaging methods such as microscopy, infrared thermography, and magnetic resonance tomography play a central role in science, technology, and medicine (*Gomès, Assy & Chapuis, 2015*; *Doi, 2006*).

Mass spectrometry imaging (MSI) datasets contain spatially resolved spectral information. The parallel detection of multiple compounds with high sensitivity established mass spectrometry (MS) as the first-choice tool for exploratory studies, particularly in combination with data mining methods (*López-Fernández et al., 2015*; *Winkler, 2015*). Numerous MSI technologies have been reported; the most commonly used MSI platform is based on matrix-assisted laser desorption/ionization (MALDI), suitable for a wide range of molecules (*Rae Buchberger et al., 2018*). However, conventional MSI

usually requires significant sample preparation and physical conditions, which are incompatible with life. Therefore, there is a keen interest in developing ambient ionization MSI (AIMSI) methods because they enable the direct analysis of delicate materials and biological tissues with no or minimal sample preparation (*Lu et al., 2018*).

Unlike digital image acquisition equipment where the luminous intensity is digitalized, MSI signal intensities must be converted to discrete values by a process termed quantization. Each discrete intensity level is known as gray level; intensity quantized images are called grayscale images. Zones in MSI spectra with distinct intensities indicate localized biochemical activity, such as the biosynthesis of natural products or physiological processes. Such *regions of interest* (ROI) can be identified by visual inspection or automated segmentation (*Gormanns et al., 2012*; *Bemis et al., 2015*).

The human vision is limited to the perception of only 700 to 900 shades of gray (*Kimpe & Tuytschaever, 2007*). With the mapping of gray levels to colors, the visualization of features in an image can be improved. However, color perception is a subjective experience affected by illumination and the individual response of rod and cone photoreceptors located in our eyes. Thus, the used color schemes significantly affect the human perception of scientific data (*Rogowitz, Treinish & Bryson, 1996*; *Race & Bunch, 2015*). The frequently used rainbow color map generates colorful images that accentuate differences in signal intensities. However, the resulting images do not comply with 'perceptual ordering,' i.e., the spectators of a rainbow-colored MSI visualization cannot intuitively assign the different colors according to the signal intensities. Thus, rainbow color schemes are confusing and even might actively mislead viewers (*Borland & Taylor li, 2007*). The online tool hclwizard (http://hclwizard.org) helps in the generation of custom color maps, which follow the hue-chroma-luminance (HCL) concept and are suitable for different types of scientific data visualizations (*Zeileis, Hornik & Murrell, 2009*; *Stauffer et al., 2015*; *Gamboa-Becerra et al., 2015*). The current gold standard for plotting scientific data is the color map Viridis, which provides linear perception and considers viewers with color vision deficiencies (CVD) (*Nuñez, Anderton & Renslow, 2018*).

Another critical aspect of image processing is the adaption of quantitative data to the human vision, i.e., an adjustment of physical data to the biological receiver (*Stockham, 1972*). Histogram equalization is widely used for enhancing the contrast in images and therefore supporting the recognition of patterns. The application of histogram equalization is simple and computationally fast. In the global histogram equalization (GHE), the entire image data are used to remap the representation levels (*Abdullah-Al-Wadud et al., 2007*). But the original intensity levels of the pixels are lost, and the fidelity of the data visualization is infringed.

Thus, we introduce the use of *Threshold Intensity Quantization* (TrIQ) for the processing of conventional and ambient ionization mass spectrometry imaging (MSI/AIMSI) datasets.

## MATERIALS AND METHODS

### Data sets and formats

We analyzed publicly available mass spectrometry imaging data sets from different MSI acquisition techniques, lateral resolutions, and sample types. The datasets used in this work are listed in Table 1.

**Table 1 Mass spectrometry imaging data sets.** AP-MALDI-Atmospheric pressure matrix assisted laser desorption/ionization, DESI-Desorption electrospray ionization, LAESI-Laser ablation/electrospray ionization, LTP-Low-temperature plasma, Res.-lateral resolution, Tol.-mass tolerance. References: *Oetjen et al. (2015)*, *Zheng, Bartels & Svatoš (2020)*, *Maldonado-Torres et al. (2014)*, *Maldonado-Torres et al. (2017)*, *Römpp et al. (2010)*, *Römpp et al. (2014)*.

| Organism, tissue, ref. | Method | Res. [$\mu$m] | Tol. [ $m/z$] | Size [pix.] |
|---|---|---|---|---|
| Human, colorectal cancer (*Oetjen et al., 2015*) | DESI (−) | 100 | ±0.3 | 67 × 64 |
| *A. thaliana*, leaf (*Zheng, Bartels & Svatoš, 2020*) | LAESI (−) | 200 | ±0.3 | 46 × 26 |
| Chili, fruit (*Maldonado-Torres et al., 2014*; *Maldonado-Torres et al., 2017*) | LTP (+) | 1,000 | ±0.3 | 85 × 50 |
| Mouse, urinary bladder (*Römpp et al., 2010*; *Römpp et al., 2014*) | AP-MALDI (+) | 10 | ±0.1 | 260 × 134 |

The original DESI data set consists of four samples per image, which we separated for further processing.

All datasets comply with the imzML data format community standard (*Schramm et al., 2012*; *Römpp et al., 2011*), implemented in many proprietary and open source programs for MSI data processing (*Weiskirchen et al., 2019*).

**Threshold intensity quantization (TrIQ) algorithm**

An image is a function $f(x, y)$ that assigns an intensity level for each point $x, y$ in a two-dimension space. For visualizing $f(x, y)$ on a computer screen or printer, the image must be digitized for both intensity and spatial coordinates. As MSI is a scanning technique, spatial coordinates $x$ and $y$ are already discrete values related to the lateral resolution of the scanning device. The intensity values provided by the MS ion detector are analog quantities that must be transformed into discrete ones. Quantization is a process for mapping a range of analog intensity values to a single discrete value, known as a *gray level*. *Zero-memory* is a widely used quantization method. The zero-memory quantizer computes equally spaced intensity bins of width $w$:

$$w = \left\lceil \frac{max(f) - min(f)}{n} \right\rceil \tag{1}$$

where $n$ represents the number of discrete values, usually 256; $min(f)$ and $max(f)$ operators provide minimum and maximum intensity values. Quantization is based on a comparison with the *transition levels* $t_k$:

$$t_k = w + min(f), 2w + min(f), \ldots, nw + min(f) \tag{2}$$

Finally, the discrete mapped value $Q$ is obtained:

$$Q(f(x, y)) = \begin{cases} 0, & f(x, y) \leq t_1 \\ k, & t_k < f(x, y) \leq t_{k+1} \end{cases} \tag{3}$$

(AI)MSI methods often produce *outliers*, i.e., infrequent extreme intensity values, which drastically reduce image contrast. The *Threshold Intensity Quantization*, or TrIQ, addresses this issue by setting a new upper limit $T$; intensities above this threshold will be

grouped within the highest bin. $T$ computation involves the *cumulative distributive function* $p(k)$ (CDF), defined as

$$q \approx p(k) = \frac{\sum_{i=1}^{k} h(i)}{N} \tag{4}$$

$h(i)$ stands for the $i$ bin's frequency within an image histogram, $N$ is the image's pixel count. Given a target probability $q$ it is possible to find the bin $k$ whose CDF closely resembles $q$. Then, the upper limit of the bin $k$ in $h$ will be used as the threshold value $T$. The new transition levels can be defined as:

$$w = \left[ \frac{T - min(f)}{n - 1} \right] \tag{5}$$

$$t_k = w + min(f), 2w + min(f), \ldots, (n-1)w + min(f) \tag{6}$$

Therefore, $Q$ mapping will be

$$Q(f(x,y)) = \begin{cases} 0, & f(x,y) \leq t_1 \\ k, & t_k < f(x,y) \leq t_{k+1} \\ n-1, & f(x,y) > t_{n-1} \end{cases} \tag{7}$$

with $k$ running from 1 to $n-1$. From Eq. (7) follows that a higher $k$ leads to a better approximation of $q$. Default values for $k$ and $q$ in RmsiGUI are 100 and 98%, respectively.

Using the default values of the TrIQ, 98% ($= q$) of the image's total intensity are visualized. Pixel intensities above the calculated threshold value $T$ are limited to a maximum value. Therefore, the rescaled 100 ($= k$) bins visualize the dataset's intensity levels with more detail.

## Implementation

Several programs and workflows for MSI data analysis employ the statistical language R (*R Core Team, 2018*), such as MSI.R (*Gamboa-Becerra et al., 2015*), Cardinal (*Bemis et al., 2015*), and the Galaxy MSI module (*Föll et al., 2019*). The Otsu segmentation method (*Otsu, 1979*) tested in this work comes with the R package EBImage (*Pau et al., 2010*). Recently, we published an R-based platform for MSI data processing with a graphical user interface, RmsiGUI, which provides modules for the control of an open hardware imaging robot (*Open LabBot*), the processing of raw data, and the analysis of MSI data (*Rosas-Román et al., 2020*). We integrated the TrIQ algorithm into RmsiGUI and provide the R code snippets for facilitating its adoption into other programs. The source code is freely available from the project repository https://bitbucket.org/lababi/rmsigui/.

We use the viridis color map, which is optimized for human perception and people with color vision deficiencies (*Nuñez, Anderton & Renslow, 2018*; *Garnier et al., 2018*). Reading and processing MSI data in imzML format are done with the MALDIquantForeign and MALDIquant libraries (*Gibb & Strimmer, 2012*; *Gibb & Franceschi, 2019*).

Figure 1 shows the graphical user interface of RmsiGUI with the TrIQ option selector.

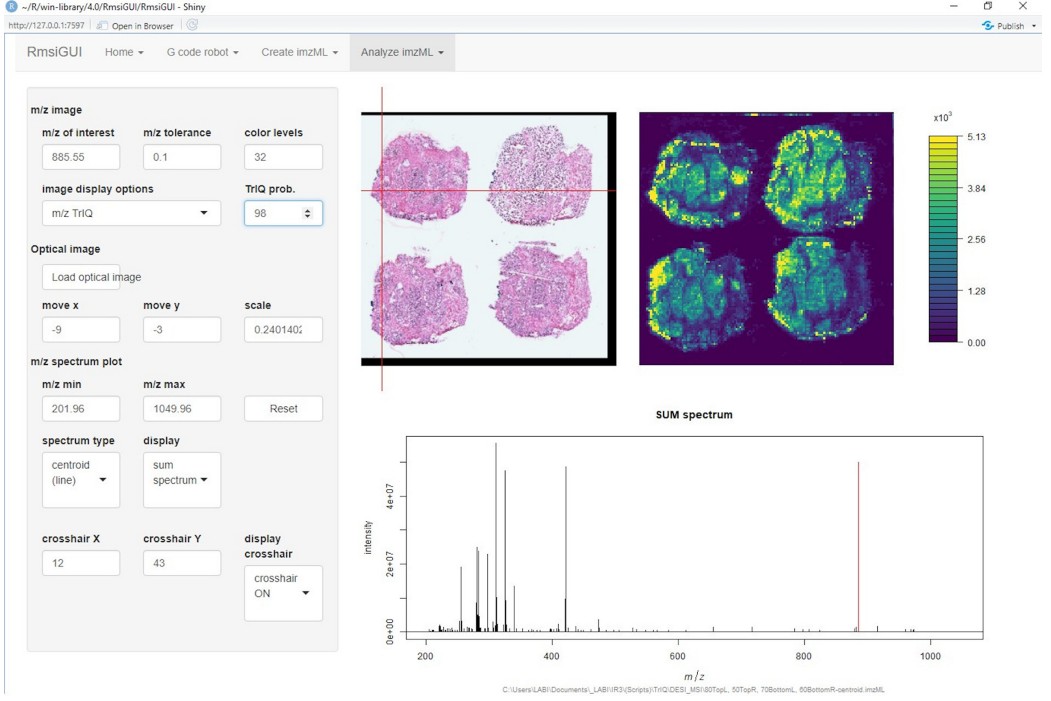

**Figure 1 Implementation of the Threshold Intensity Quantization (TrIQ) in the graphical user interface of RmsiGUI (dataset and image from *Oetjen et al. (2015)*).**

# RESULTS AND DISCUSSION

The contrast of digital images can be enhanced with additional operations, such as global or local histogram equalization algorithms; however, such image processing tools do not preserve the original gray level scale's linearity. In contrast, our *Threshold Intensity Quantization* (TrIQ) approach finds an intensity threshold $T$ for saturating the images' last gray level. Importantly, the linearity of the experimentally determined intensity scale is preserved.

In the next sections, we demonstrate the application of the TrIQ for the processing of mass spectrometry imaging (MSI) datasets. The term *raw image* is used in this paper for denoting images rendered with the default quantization method of R.

## Contrast optimization

Figure 2 shows the mass spectrometry image of a human colorectal adenocarcinoma sample, acquired with DESI and 100 $\mu$m spatial resolution (*Oetjen et al., 2015*). The imaged signal of 885.55 *m/z* corresponds to de-protonated phosphatidylinositol (18:0/20:4), $[C_{47}H_{83}O_{13}P\text{-}H]^-$ (*Tillner et al., 2017*).

The direct plotting of the extracted *m/z* slice results in the image shown in Fig. 2A. The image contains pixels with intensity values of up to 70,280 arbitrary units. Such extreme and infrequent intensity values are called *outliers* and drastically reduce image contrast. The histogram below reveals the reason for the low contrast of the image: Most of the pixels fall into the first four bins after the default R quantization.

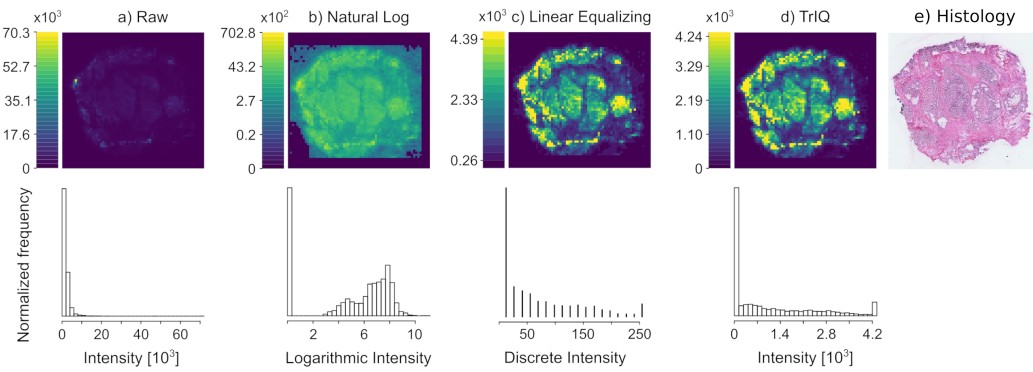

**Figure 2 Contrast enhancement by Threshold Intensity Quantization (TrIQ).** The mass trace 885.55 *m/z* of DESI MSI data of a human colorectal adenocarcinoma sample was visualized with different quantization methods. (A) Raw data, rendered with 256 gray levels. (B) Plotting with applying the natural logarithm to the intensity values. (C) Image contrast improvement after zero-memory quantization and linear histogram equalization (D) Image contrast improvement using TrIQ with 95 percent and 32 gray levels. Histograms A, B and D were computed with 32 bins; histogram C has 256 bins. (E) Histological staining of a tissue slice (dataset and image E) from *Oetjen et al. (2015)*.

A typical data transformation for imaging is the use of a logarithmic intensity scale. Figure 2B shows the image after applying the natural logarithm to the MSI signal intensities and default quantization. The contrast is improved. However, further operations would be necessary, such as the subtraction of the background level. Besides, the interpretation of the non-linear color scale is not intuitive.

The conventional sequence for improving the contrast of an MSI image is applying the zero-memory quantization of MSI data and a transformation function on the quantized pixels. Figure 2C shows the result of this process. Although improved contrast is gained with linear equalization, the corresponding histogram shows that many gray levels were lost in the transformation process. Only eighteen remaining gray levels contain data. This information loss is irreversible and may have adverse effects on the performance of standard segmentation algorithms. To make TrIQ and equalized images comparable, we selected an intensity threshold for saturating the equalized image near to the $T$ threshold provided by the TrIQ algorithm. The shown linear equalization histogram has 256 bins.

Applying the TrIQ algorithm with $P = 0.95$ resulted in Fig. 2D. Pixels with intensities above $T = 4,244$ were binned within the highest gray level. The corresponding histogram shows an almost flat frequency over a wide intensity range. Compared to the unprocessed image, the contrast of the image was drastically improved, and the linear intensity scale of the color representation was preserved. Figure 2E shows the histological staining of the tissue section analyzed. The mass image processed with the TrIQ algorithm reassembles best the anatomical details which are recognizable in the optical image.

## Background optimization

Various causes can result in background noise, such as sample metabolites with low vapor pressure and matrix compounds. Figure 3A shows the rendering of signals from an

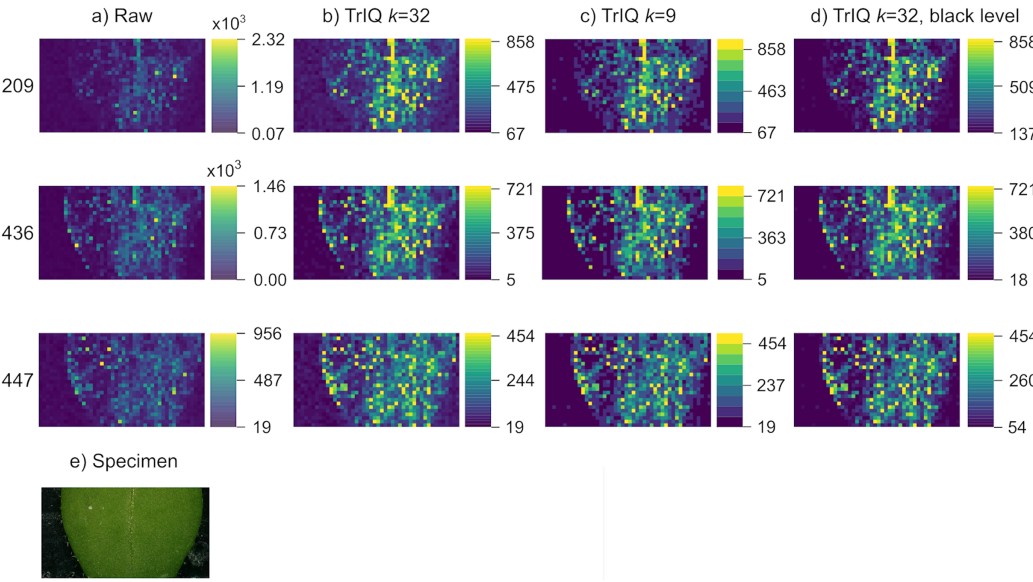

**Figure 3 Background optimization with TrIQ.** Three mass traces of LAESI MSI data from a *A. thaliana* leaf are plotted. (A) Raw data plotting with 256 gray levels. (B) TrIQ with 32 gray levels. (C) TrIQ with 9 gray levels. The background uniformity is improved by reducing the intensity and gray levels. (D) TrIQ with 32 gray levels and black level adjustment eliminates background noise without reducing color depth (dataset from *Zheng, Bartels & Svatoš (2020)*). (E) Photograph of the sample.

*Arabidopsis thaliana* leaf, analyzed by LAESI MSI with a lateral resolution of 200 $\mu$m (*Zheng, Bartels & Svatoš, 2020*). The images correspond to the putative negative ions of 4-hydroxymethyl-3-methoxyphenoxyacetic acid ($[C_{10}H_{12}O_5\text{-}3H]^-$, 209.0 *m/z*), 4-methylsulfonylbutyl glucosinolate ($[C_{12}H_{23}NO_{10}S_3\text{-}H]^-$, 436.0 *m/z*), and indol-3-ylmethyl glucosinolate ($[C_{16}H_{20}N_2O_9S_2\text{-}H]^-$, 447.1 *m/z*) (*Wu et al., 2018*).

Removing outliers with TrIQ and reducing gray levels lead to improved image brightness (see Figs. 3A and 3B). Nevertheless, background noise is also enhanced, and the sample shape is not well defined (see ion 209 *m/z* in Fig. 3B). There are two possibilities for background correction with TrIQ. The first option is reducing the number of gray levels, thus grouping a wider range of values within a single bin. Reducing the gray levels from 32 to 9 produced an almost perfectly uniform background and a well-defined sample shape, as shown in Fig. 3C. The second method finds a new *black level threshold* that substitutes the operator $min(f)$ in Eq. (5). As the color bars for this approach indicate, the black level thresholds depend on the individual image data (see Fig. 3D). Both methods efficiently diminish the impact of background noise. But whereas reducing the gray levels is the simpler approach, defining a new black level threshold maintains the color depth.

## Normalization for comparable mass spectrometry images

Comparing mass spectrometry images (MSI) is a challenge because standard quantization procedures create images with distinct intensity and color scales, even if they were measured under the same experimental conditions.

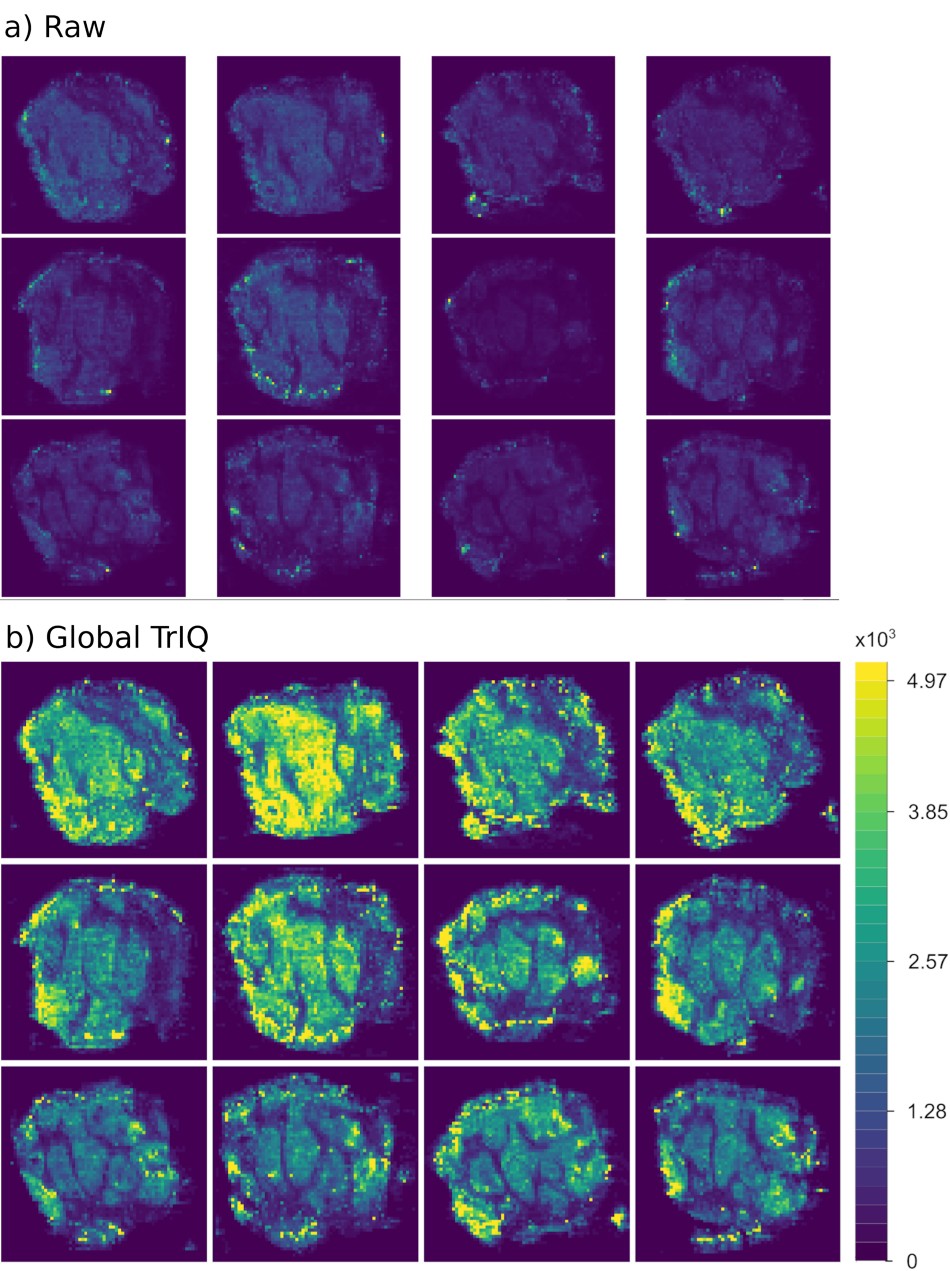

**Figure 4 Global TrIQ applied to the ion 885.55 *m/z* of human colorectal adenocarcinoma DESI MSI slices, with *P* = 0.91 and 32 gray levels.** Compared to the raw data visualization (A), the contrast is drastically enhanced by applying the TrIQ algorithm (B). The normalization also allows a direct comparison of the images (dataset from *Oetjen et al. (2015)*).

Global TrIQ finds the highest $T$ among a given MSI set. This threshold is used for computing the transition levels and mapping MSI intensities to discrete values on every image within the set.

Figure 4 shows human colorectal adenocarcinoma images. The samples come from the same tissue, cut into slices with a thickness of 10 $\mu$m (*Oetjen et al., 2015*). All images visualize the abundance of the ion 885.55 *m/z*. The plotting of the raw data resulted in

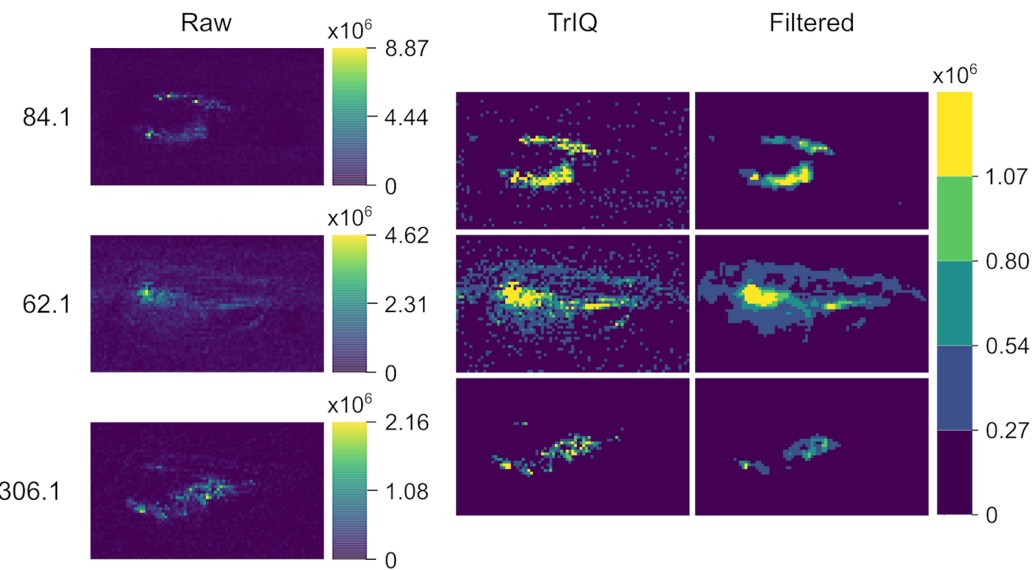

**Figure 5 LTP MSI of a chili _C. annuum_ fruit with 1 mm lateral resolution.** TrIQ with $P = 0.98$ and 5 gray levels improves the contrast; an additional median filter removes technical noise (dataset from _Maldonado-Torres et al. (2014, 2017)_).

images of low contrast, which are difficult to interpret and to compare (see Fig. 4A). Using the Global TrIQ algorithm, a maximum threshold $T$ of 4,970 was calculated and applied for all twelve images; the accumulated probability was set to 0.91. The resulting color scale is the same for all images (see Fig. 4B). Thus, the relative abundance and distribution of the ion in multiple images can be evaluated at a single glance.

Figure 5 provides another example of global TrIQ. The image compares the 62.1, 84.1, and 306.1 _m/z_ ions of chili (_Capsicum annuum)_ slices sampled at 1 mm lateral resolution with low-temperature plasma (LTP) MSI (_Maldonado-Torres et al., 2014_). LTP MSI detects small, volatile compounds. Therefore, the images resulting from this ambient ionization technique are noisy. TrIQ with $P = 0.98$ and five gray levels improves the contrast. The remaining noise is efficiently removed with a $3 \times 3$ median filter (see Fig. 5). The signals with a mass-to-charge ratio of 62.1 and 84.1 display a defined localization at the seeds and at the center of the placenta. The signal with 306.1 _m/z_ is enriched in placenta tissue and corresponds to the $[C_{18}H_{27}NO_3+H]^+$ ion of capsaicin, the compound responsible for the hot taste of chili peppers (_Aza-González, Núñez-Palenius & Ochoa-Alejo, 2011_; _Maldonado-Torres et al., 2014_; _Gamboa-Becerra et al., 2015_; _Cervantes-Hernández et al., 2019_).

Both examples demonstrate the usefulness of TrIQ to normalize MSI data for image comparison.

## Segmentation and visualization of regions of interest (ROI)

Finding regions of interest (ROI) in MSI datasets is essential for biomarker discovery and physiological studies. High-contrast and low-noise images favor the automated segmentation. Figure 6 compares binary masks obtained using the standard Otsu algorithm and TrIQ. The input image was zero-memory quantized with 256 bins before

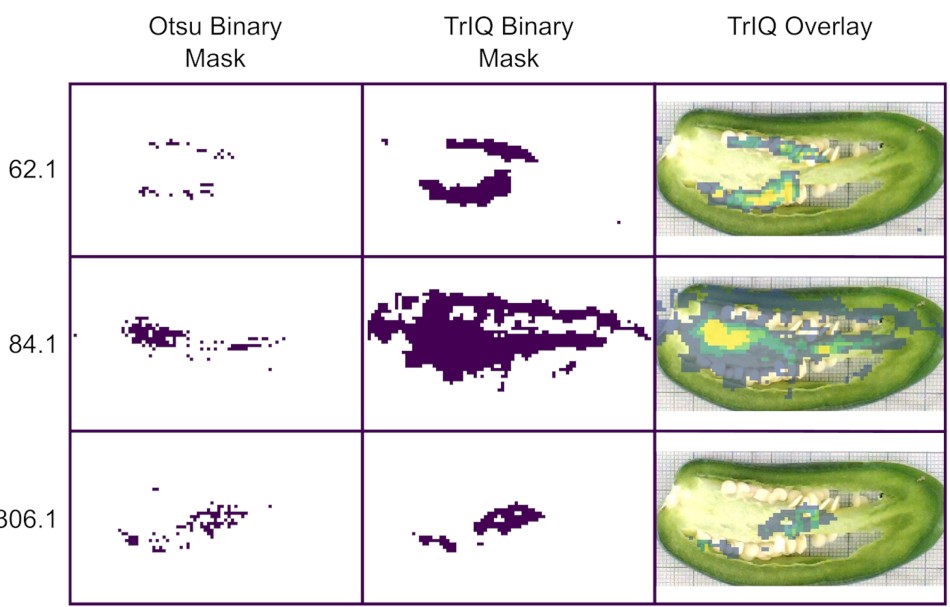

**Figure 6  Image segmentation for *C. annuum* obtained with Otsu and TrIQ.** For images with intensity outliers and noisy regions, TrIQ combined with median filtering gives larger and more uniform regions compared to the standard Otsu algorithm (dataset from *Maldonado-Torres et al. (2014*, *2017)*).

applying the Otsu algorithm. The TrIQ binary masks were obtained by setting TRUE all non-zero values of the median filtered images of Fig. 5.

TrIQ-median filtered images show more extensive and homogenous regions compared to the Otsu masks and fit well with the anatomical fruit sections of the optical image (see Fig. 6).

Figure 7 shows the Global TrIQ segmentation of mouse urinary bladder data, scanned with AP-MALDI MSI at a lateral resolution of 10 $\mu$m (*Römpp et al., 2010*; *Römpp et al., 2014*). The distribution of 741.53 *m/z*, identified as a sphingomyelin [$C_{39}H_{79}N_2O_6P$ + K]$^+$ ion, is related to muscle tissue. 743.54 *m/z* is associated with the *lamina propria* structure, while 798.54 *m/z* is mainly found in the *urothelium* (*Römpp et al., 2010*). Global TrIQ was applied with $P = 0.95$ and 25 gray levels. TrIQ and median filtering allow clear discrimination between muscle tissue and *lamina propria* using the marker ions 741.53 and 743.54 *m/z*. For separating the *urothelium* structure from the muscular tissue, the binary image for 798.54 *m/z* was calculated by zeroing gray levels below nine. In contrast, if the Otsu method is applied to the 798.54 *m/z* ion, the *urothelium* region is isolated automatically. This result is expected since the Otsu method assumes an image histogram distribution with a deep sharp valley between two peaks. Figure 8 shows an overlay of the TrIQ processed ion images, representing correctly the anatomical structures of the mouse urinary bladder.

## Computational performance of the algorithm

For estimating the computational performance of the TrIQ algorithm, we reprocessed selected ion traces presented in this paper (DESI 885.55 *m/z*, LAESI 209 *m/z*, LTP 62.1 *m/z*,

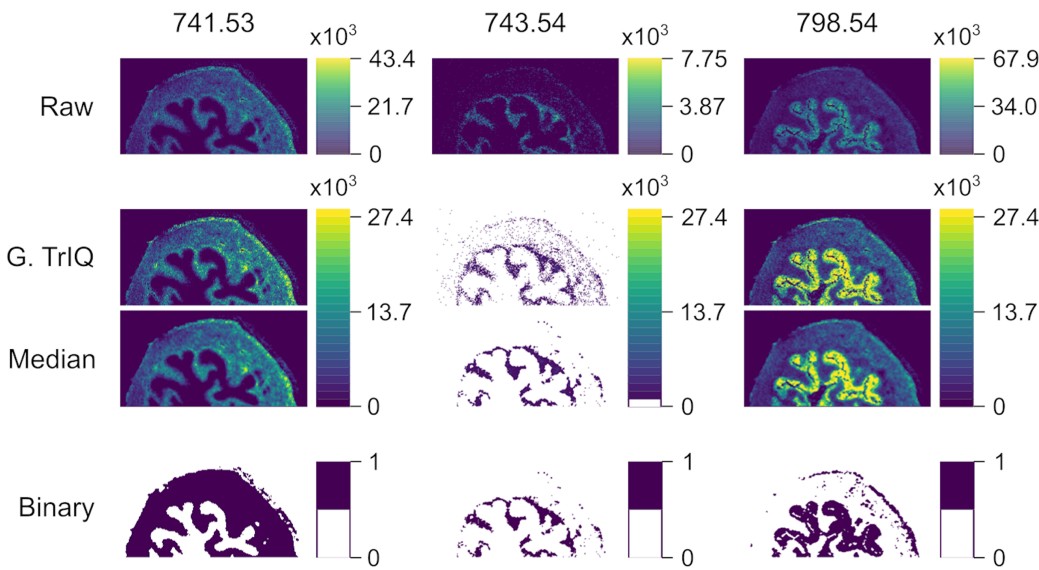

**Figure 7 AP-MALDI MSI of a mouse urinary bladder imaged with a lateral resolution of 10 μm.**
TrIQ was applied with *P* = 0.95 and 25 gray levels. Binary images serve for defining regions of interest
(ROI) and segmentation: 741.53 *m/z*-muscle tissue, 743.54 *m/z*-lamina propria structure, 798.54 *m/z*-
urothelium (dataset from *Römpp et al. (2010, 2014)*).      

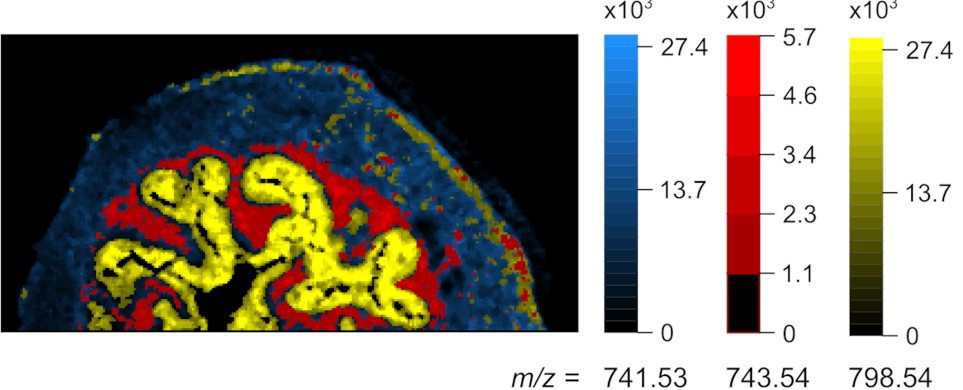

**Figure 8 Overlay image, representing the anatomical structures in mouse urinary bladder.** Blue:
741.53 *m/z*-muscle tissue, red: 743.54 *m/z*-lamina propria, yellow: 798.54 *m/z*-urothelium (dataset from
*Römpp et al. (2010, 2014)*).      

and AP-MALDI 741.53 *m/z*) and measured the time for applying the TrIQ. To test the
suitability of the TrIQ algorithm for large images, we built synthetic slices by sequentially
doubling the DESI data. Time calculation was executed 25 times to account for variations
in system processes of the operating system. Figure 9 demonstrates the results from
running the R script Timing. R (provided as supplemental code) on a standard Linux
laptop (Intel(R) Core(TM) i7-7700HQ CPU with 2.80 GHz, 16 Gb RAM, Peppermint OS
10). On average, more than 1,000 pixels were processed per millisecond. In the tested
range, i.e., up to >500,000 pixels, the execution speed was proportional to the image's size.
Thus, the TrIQ algorithm is computationally efficient and scalable.

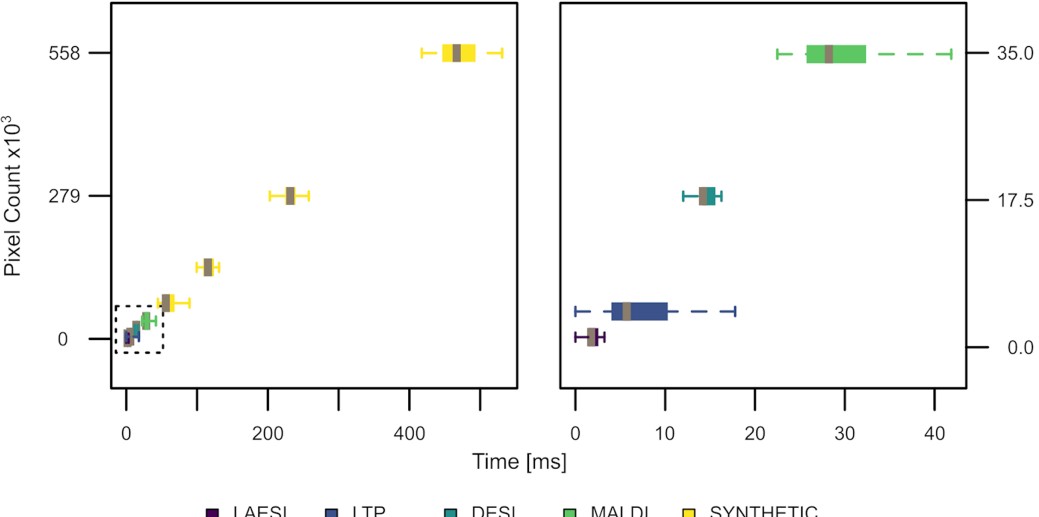

**Figure 9 Execution speed of the TrIQ algorithm, implemented in R, using a standard laptop (experimental and synthetic data).** About 1,000 pixels were processed per millisecond. The TrIQ algorithm scales linearly.                                     

## CONCLUSIONS

Threshold Intensity Optimization (TrIQ) improves the visualization of mass spectrometry imaging (MSI) data by augmenting the contrast and homogenizing the background. Contrary to histogram equalization algorithms, TrIQ preserves the linearity of measured ion intensities.

The processing with TrIQ facilitates the recognition of regions of interest (ROI) in MSI data sets, either by visual inspection of by automated segmentation algorithms, supporting the interpretation of MSI data in biology and medicine.

As with any data filtering method, TrIQ could remove or alter valuable information. Therefore, it is recommended to compare raw and processed images and carefully adjust the target probability and the number of gray levels. MSI technology itself causes further sources of errors. Fixation agents, solvents, and matrix compounds can lead to background signals. To some extent, such off-sample ions can be removed manually, or using MSI software (*Ovchinnikova et al., 2020*). Technical variations caused by the sample topology or unstable ionization add additional inaccuracies (*Bartels et al., 2017*). Further, the ion count is not strictly proportional to the abundance of a molecule but depends on the local sample structure and composition, and the desorption/ionization principle. Thus, complementary MSI, optical and histological methods should be used on the same sample (*Swales et al., 2019*).

Applying TrIQ to a set of images equalizes their intensity scales and makes them comparable. We demonstrated the implementation of the TrIQ algorithms in R to process MSI data in the community format imzML. The algorithm is computationally fast and only requires basic operations, and thus can be quickly adapted to any programming language. TrIQ can be applied to improve any scientific data plotting with extreme values, respecting the original intensity levels of raw data.

## CODE AVAILABILITY

RmsiGUI is freely available from https://bitbucket.org/lababi/rmsigui/. We released TrIQ R scripts and the R package RmsiGUI under the terms of the GNU General Public License, GPL V3 (http://gplv3.fsf.org/).

## ACKNOWLEDGEMENTS

We thank Dr. Abigail Moreno Pedraza, Dr. Teresa Maria Teresa Carrillo Rayas, Maria Isabel Cristina Elizarraraz Anaya, and Thermo Mexico for technical support and the research groups that made their code and MSI datasets public.

### Funding

The project was funded by the CONACyT bilateral grant CONACyT-DFG 2016/277850. Ignacio Rosas-Román received a CONACyT PostDoc scholarship. The funders had no role in study design, data collection and analysis, decision to publish, or preparation of the manuscript.

### Grant Disclosures

The following grant information was disclosed by the authors:
CONACyT bilateral grant: CONACyT-DFG 2016/277850.
CONACyT PostDoc scholarship.

### Competing Interests

Robert Winkler is an Academic Editor for PeerJ, co-inventor of the patent application "Non-thermal plasma jet device as source of spatial ionisation for ambient mass spectrometry and method of application" (WO 2014/057409) and a shareholder of the company Kuturabi SA. de CV.

### Author Contributions

- Ignacio Rosas-Román conceived and designed the experiments, performed the experiments, analyzed the data, performed the computation work, prepared figures and/or tables, authored or reviewed drafts of the paper, and approved the final draft.
- Robert Winkler conceived and designed the experiments, prepared figures and/or tables, authored or reviewed drafts of the paper, and approved the final draft.

### Patent Disclosures

The following patent dependencies were disclosed by the authors:
  Robert Winkler is a co-inventor of the patent application "Non-thermal plasma jet device as source of spatial ionisation for ambient mass spectrometry and method of application" (WO 2014/057409).

## Data Availability

The software RmsiGUI is freely available from https://bitbucket.org/lababi/rmsigui/.

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
