# Peer review of "Contrast optimization of mass spectrometry imaging (MSI) data visualization by threshold intensity quantization (TrIQ)"

_PeerJ Computer Science, doi:10.7717/peerj-cs.585_

## Round 0.1 · original submission · Major Revisions

Please address the reviewers' comments point by point and revise the articles accordingly.

Reviewer 1 ·

Basic reporting

No remarks on the basic reporting, only a suggestion for figure 9.
Figure 9:
It is unfortunate that the majority of the information related to (LAESI, LTP, DESI, MALDI) shown in this graph is crowded in the figure, making it difficult to interpret this information. Perhaps it would help to adapt the scale to improve readability of this figure?

A small correction:

Line 232: Figure 7 -> Figure 8

Experimental design

Figure 2 and lines 165 - 166 ‘The mass image processed with the TrIQ algorithm reassembles best the anatomical details which are recognizable in the optical image’

The contrast enhancement difference between using the natural logarithmic versus a linear equalizing approach comes out clearly. It could be due to the quality of the images but currently it is not easy to see where TriQ improves over the Linear Equalizing approach. Which anatomical details are you referring to? It would be good to clarify or highlight the obtained improvements and in particular the anatomical details you are referring to in the images.

Figure 3 – background optimization
How does the TriQ approach compare to the state-of-the art for background optimization in addition to the raw results shown? It would also be good to add the histology section for comparison as well.

Figure 4 – lines 193 – 198 Global TriQ
How do the results shown in figure 4 compare to the slices without global TriQ being applied?

Validity of the findings

I like the approach of the global TriQ to improve comparability of ion images in MSI data, which is an important problem to solve. However, I feel the appropriate controls are missing making it difficult to properly assess the validity of the findings.

·

Basic reporting

The manuscript descirbes the implementation of a Threshold Intensity Quantization algorithm for augmenting the contrast in Mass Spectrometry Imaging (MSI) data visualizations.
The authors method provides an improvement of MSI data through increasing the contrast and homogenizing the background.

The language used throughout the manuscript is clear.

Experimental design

The method was implemented as an open-source software RmsiGUI with an R script for post-processing.
They validated the developed method using different dataset acquired from different techniques. The results show an improvement in the contrast and in the detection of ‘region of interest’.

I compliment the authors on their vast data set used to validate the method. However, the mathematical part in the section 2.2 need to be improved by providing more details.
The authors claimed that their algorithm is computationally fast (line 266) but no strong information is provided to justify this. Please provide a quantifiable evidence (running time, memory expenses, …) to justify this, since figure 9 is not clear and does not provide sufficient details.

Validity of the findings

The research question is well defined.
The authors stated that several programs for MSI data analysis exists and employ the statistical language R but did not compare their developed algorithm with these existing ones. Authors are suggested to include one or more of these existing methods as a comparison to prove that their method is working fine.
Authors are also suggested to provide proper applications of this tools as the novelty and the impact is not well assessed.

Additional comments

The introduction shows the context but needs more details. I suggest that you improve the description by some up-to-date references.
The figures are relevant and well labeled, however some are not of high quality (Figure 1 and Figure 9).

---

## Round 0.2 · Minor Revisions

Please see the remaining comments from Reviewer 2 and provide rebuttal details.

Reviewer 1 ·

Basic reporting

No further comments.

Experimental design

No further comments.

Validity of the findings

No further comments.

Additional comments

Nice work!

·

Basic reporting

The manuscript descirbes the implementation of a Threshold Intensity Quantization algorithm for augmenting the contrast in Mass Spectrometry Imaging (MSI) data visualizations.
The authors method provides an improvement of MSI data through increasing the contrast and homogenizing the background.
The method was implemented as an open-source software RmsiGUI with an R script for post-processing.
They validated the developed method using different dataset acquired from different techniques.
The results show an improvement in the contrast and in the detection of ‘region of interest’.
The language used throughout the manuscript is clear.

Experimental design

The introduction shows the context but needs more details. I suggest that you improve the
description by some up-to-date references.
The figures are relevant and well labeled, however some are not of high quality (Figure 1 and Figure 9).
The research question is well defined.
I compliment the authors on their vast data set used to validate the method. However, the mathematical part in the section 2.2 need to be improved by providing more details.
The authors claimed that their algorithm is computationally fast (line 266) but no strong information is provided to justify this. Please provide a quantifiable evidence (running time, memory expenses, …) to justify this, since figure 9 is not clear and does not provide sufficient details.
The authors stated that several programs for MSI data analysis exists and employ the statistical language R but did not compare their developed algorithm with these existing ones.

Validity of the findings

Authors are suggested to include one or more of these existing methods as a comparison to prove that their method is working fine. Authors are also suggested to provide proper applications of this tools as the novelty and the impact
is not well assessed.

---

## Round 0.3 · accepted · Accept

The authors have addressed all the comments from the reviewer.